# The Influence of Camera Calibration on Nearshore Bathymetry Estimation from UAV Videos

Gonzalo Simarro [1,*,†], Daniel Calvete [2,†], Theocharis A. Plomaritis [3], Francesc Moreno-Noguer [4], Ifigeneia Giannoukakou-Leontsini [1,3], Juan Montes [5] and Ruth Durán [1]

1   Instituto de Ciencias del Mar (ICM, CSIC), Passeig Marítim de la Barceloneta 37-49, 08003 Barcelona, Spain; i.giannoukakouleonts@studio.unibo.it (I.G.-L.); rduran@icm.csic.es (R.D.)
2   Department of Physics, Universitat Politècnica de Catalunya, Jordi Girona 1-3, 08034 Barcelona, Spain; daniel.calvete@upc.edu
3   Department of Applied Physics, Instituto Universitario de Investigación Marina (INMAR), Campus de Excelencia Internacional/Global del Mar (CEI-MAR), University of Cadiz, 11510 Puerto Real, Spain; haris.plomaritis@uca.es
4   Institut de Robòtica i Informàtica Industrial (IRI, CSIC-UPC), Llorens i Artigas 4-6, 08028 Barcelona, Spain; fmoreno@iri.upc.edu
5   Department of Earth Science, Instituto Universitario de Investigación Marina (INMAR), Campus de Excelencia Internacional/Global del Mar (CEI-MAR), University of Cádiz, 11510 Puerto Real, Spain; juan.montes@uca.es
*   Correspondence: simarro@icm.csic.es
†   These authors contributed equally to this work.

**Abstract:** Measuring the nearshore bathymetry is critical in coastal management and morphodynamic studies. The recent advent of Unmanned Aerial Vehicles (UAVs), in combination with coastal video monitoring techniques, allows for an alternative and low cost evaluation of the nearshore bathymetry. Camera calibration and stabilization is a critical issue in bathymetry estimation from video systems. This work introduces a new methodology in order to obtain such bathymetries, and it compares the results to echo-sounder ground truth data. The goal is to gain a better understanding on the influence of the camera calibration and stabilization on the inferred bathymetry. The results show how the proposed methodology allows for accurate evaluations of the bathymetry, with overall root mean square errors in the order of 40 cm. It is shown that the intrinsic calibration of the camera, related to the lens distortion, is the most critical aspect. Here, the intrinsic calibration that was obtained directly during the flight yields the best results.

**Keywords:** Unmanned Aerial Vehicles (UAVs); camera calibration; bathymetry estimation; coastal morphodynamics; coastal management



## 1. Introduction

Measuring the nearshore bathymetry is a fundamental challenge in coastal zone management [1–4]. Accurate bathymetries allow for subsequent decision making (e.g., whether or not it is necessary to dredge the mouth of a harbour). When several bathymetries are available over time, the seaward limit of significant sediment cross-shore transport can be identified [5], enabling the study of shoreface morphodynamics [6], the estimation of sediment budget and pathways [7], and the validation of morphodynamic models, which, in turn, are helpful in predicting future changes [8,9].

While in situ techniques (e.g., [1,10]) can provide excellent bathymetries, they are expensive, highly time consuming, and weather restricted. Remote sensing techniques can be used as an alternative, as they can collect image data of the sea surface, where bathymetry can be retrieved against lower cost and under a wider range of sea state conditions. These techniques include LiDAR (Laser imaging, Detection And Ranging) [11], X-band radar images [12,13], and, the focus of this work, optical video images [14–17]. The

preference of one technique over the others will depend on aspects, such as the dimensions of the study site, the desired spatial and temporal resolution, the required accuracy, the predominant weather conditions, and the available budget.

Video monitoring stations [18–21], which are frequently referred to as "Argus" stations [19], were developed after the relatively recent advent of digital cameras (around 30 years ago) and they have been shown to be a very powerful and low-cost tool for long term monitoring of coastal sites, collecting imagery that covers up to one kilometer and has a time sampling frequency of one hour. Imagery has been used to analyze, e.g., shoreline variability [3,22,23], intertidal bathymetry [24,25], beach morphology [26–29] and submerged bathymetry estimation [14–17]. The recent improvement and cost reduction of small Unmanned Aerial Vehicles (UAVs) allows to use the tools developed for video monitoring stations in places where measurements are required and no video station is available, either because there is not a high vantage point or because only a single survey is needed. Recent works have already explored the use of videos recorded with UAVs to obtain bathymetry [30–32].

Nearly all of the methods that retrieve the bathymetry from video images rely on the dispersion relationship that relates the wavelength, the wave period, and the water depth. Therefore, the problem reduces to the estimation of the wave period (nearly spatially invariant) and the wavelength (spatially varying). While early methods analyzed the video following one-dimensional (1D) transects to obtain the wave period and cross-shore wavelength, more recent methods analyze the video in a two-dimensional (2D) way: "cBathy" [16] while using Fourier Analysis and "uBathy" [17] through Principal Component Analysis.

Obtaining an accurate calibration of the cameras is one of the main challenges when using videos, as small errors may propagate to large discrepancies when inferring the bathymetry [32]. In order to perform the calibration, a series of stable points that are visible in several images are required. However, these points, denoted as Ground Control Points (GCPs), are limited to the dry (and often small) area of the image, while the wet zone, where the bathymetry is to be obtained, occupies the other part of the image. This issue, which is also present in Argus-like stations, is particularly important when using UAVs, since the GCPs are often ephemeral (targets displayed on the beach).

The goal of this work is to gain a better understanding on the influence of the calibration and the stabilization of the video on the bathymetry estimation. Experimental data include an echo-sounder bathymetry performed at the same time when two UAV videos were recorded (Sections 2.1 and 2.2). Section 2.3 describes yhe frame-by-frame video calibration methodology, and the bathymetry estimation is obtained with *uBathy* [17] (Section 2.4). The results for the calibration and stabilization are presented in Section 3.1 and the final bathymetries are reported in Section 3.2. Section 3 also includes a sensitivity analysis of the camera calibration, camera stabilization, and other parameters that affect the bathymetry estimation (Sections 3.3–3.5). A short discussion of the results is included in Section 4 and, finally, the conclusions are drawn in Section 5.

## 2. Methodology

### 2.1. Study Site: Victoria Beach

Victoria Beach is a 3 km long urban beach that is situated in the Atlantic SW coast of Spain (Figure 1A), a meso-tidal and semi-diurnal environment with an average tidal range of 2 m and a mean spring tidal range that can reach up to 3 m. The wave climate in the area is characterized by an average annual offshore significant wave height ($H_s$) of 0.8 m and peak period ($T_p$) of 9 s, with waves approaching predominantly from WSW. During the storm season (from November to March), the monthly averaged values are $H_s = 1.0$ m and $T_p = 10$ s, and WSW directions dominate. During the calm season (from April to October), $H_s = 0.7$ m, $T_p = 8$ s, and Westerly wave directions dominate. The interannual wave climate is modulated by NAO (North Atlantic Oscillations) phases with higher storm frequency during negative phases [33].

Geomorphologically, Victoria Beach consists of a rectilinear and sandy beach with a SSE-NNW orientation, being backed by a promenade that delimits the sandy and urban sections of the coast. The bathymetric contours are broadly parallel to the coastline (Figure 1A). As a result of its orientation, the study area is exposed to the predominant WSW wave conditions that drive a net littoral drift towards the southeast. The beach has a dissipative to intermediate beachface with slope values that range from 0.020 to 0.025 and a dry zone with a width of approximately 60–100 m [34]. The beach is composed by medium to fine quartz-rich sands ($d_{50} \sim 0.20$ mm), and a series of rocky outcrops are present at the lower part of the beachface and upper shoreface [35]. Mesoforms have been observed in the study area, including beach cusps and intertidal flat bars. The bars have a seasonal behavior and their position is influenced by the rocky outcrop [36]. The shoreline position is stable, with long-term erosion rates below 0.75 m/yr [37].

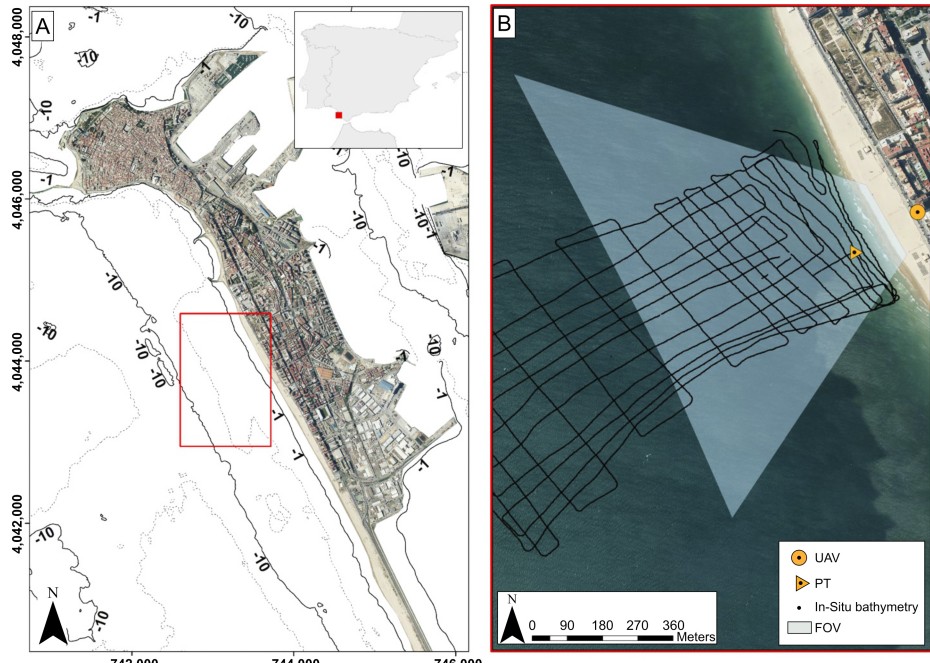

**Figure 1.** Regional map of Cádiz highlighting, with a red box, the study area of Victoria Beach (**A**); aerial image of Victoria Beach, showing the position of the drone (UAV) and corresponding (approximate) field of view (FOV), the pressure transducer position (PT), and the in situ echo-sounder track (**B**).

*2.2. Data Collection*

The field campaign was performed on 29 and 30 October 2019, including an in situ bathymetric survey, the deployment of a pressure transducer, two UAV flights, and ground control points (GCPs) acquisition. As for the in situ bathymetry measures, a single beam 235 kHz Ohmex Sonarmite v3.0 echo-sounder (Ohmex Ltd., Sway Hampshire, UK), with sample rate of 1 Hz and a theoretical vertical resolution of 0.05 m, was mounted on a RIB (Rigid Inflatable Boat), together with RTK-GPS positioning (Leica Geosystems AG, Heerbrugg, Switzerland). The bathymetric survey covered an area of approximately 400 m long-shore and 700 m cross-shore (Figure 1B, Figure 2).

The pressure transducer, deployed at the upper shoreface (Figure 1B) for 49 h covering the entire duration of the field work, was set to continuously record the surface elevation at a frequency of 4 Hz in order to obtain both tidal and wave characteristics. The tidal signal was obtained through a low-pass filter. The tidal signal from the transducer and the RTK-GCPs were used for obtaining the bathymetric data.

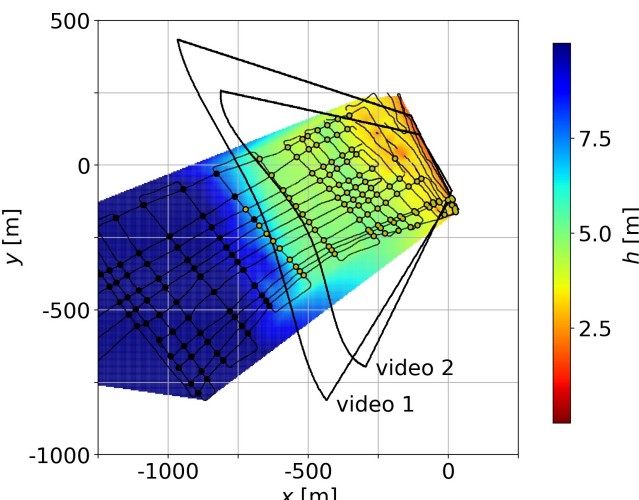

**Figure 2.** Ground truth bathymetry: path of the echo-sounder. The path intersections for $h \leqslant 7.5$ m are highlighted in yellow. The approximate domains for both videos are also included.

Figure 2 shows the echo-sounder path, together with the interpolated bathymetry. The interpolated bathymetry is presented for illustrative purposes only. The intersections of the path of the echo-sounder have been used to give an approximation of the inner error of the ground truth. For this purpose, only the intersections where the average of the two values of the water depth is $\leqslant 7.5$ m are considered. These intersections are highlighted in the figure and they correspond, roughly, to the domain for both videos. The root mean square of the difference between the water depths measured (through the two paths) at the intersections is 0.15 m. This value can be considered to be a lower limit for the errors that were obtained from the video.

Two videos, of about 10 min. each, were recorded while using a DJI Phantom 3 Pro quadcopter. The UAV was positioned at the rear end of the beach looking offshore and hovering at an approximately constant horizontal coordinates (Figure 1B) and height of ∼100 m for video 1 (Figure 3A) and ∼50 m for video 2 (Figure 3B). The initial frame resolution of the video was $4096 \times 2160$ (pixels × pixels) at a sampling rate of 24 Hz. The video frames were downsampled to $2048 \times 1080$ at 2 Hz, obtaining 1225 frames for video 1 and 1223 frames for video 2. The interval of the frame acquisition was defined based on the computational cost and the type of data that were needed to be extracted. Prior to the flights, a total of 34 GCP targets were positioned in the beach (Figure 3, not all were visible) and their coordinates were obtained with a RTK-GPS with tilt compensation (Leica GS18).

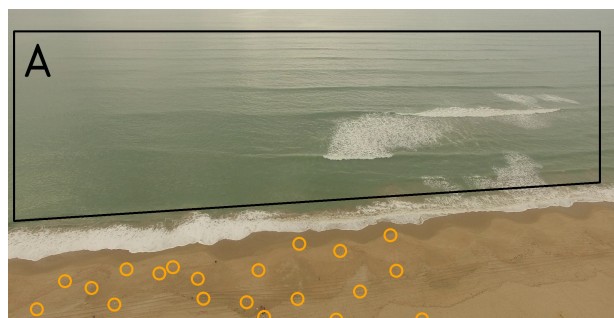
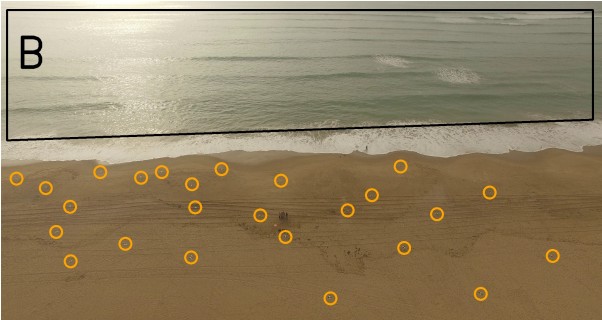

**Figure 3.** Targets displayed as georeferenced points (GCPs) for videos 1 (**A**) and 2 (**B**). The visible targets are marked with circles. The black quadrangles stand tor the pixel domains employed.

### 2.3. Video Calibration and Stabilization

### 2.3.1. Camera Model

To process video information, the basic equations of the projective geometry were first defined to map three-dimensional (3D) real-world coordinates into pixel coordinates. In an ideal situation where the camera has no distortion, the pixel coordinates, column $c_U$, and row $r_U$, for a point $\mathbf{x} = (x, y, z)$ in the real-world, are [38]

$$c_U = \frac{u_{U\star}}{s_{c\star}} + o_c, \qquad r_U = \frac{v_{U\star}}{s_{r\star}} + o_r, \tag{1}$$

where $s_{c\star}$ and $s_{r\star}$ stand for the pixel size (hereinafter "$\star$" denotes dimensionless magnitudes), $o_c$ and $o_r$, in pixels, correspond to the principal point (usually near the center of the image) and

$$u_{U\star} = \frac{(\mathbf{x} - \mathbf{x_c}) \cdot \mathbf{e_u}}{(\mathbf{x} - \mathbf{x_c}) \cdot \mathbf{e_f}}, \qquad v_{U\star} = \frac{(\mathbf{x} - \mathbf{x_c}) \cdot \mathbf{e_v}}{(\mathbf{x} - \mathbf{x_c}) \cdot \mathbf{e_f}}, \tag{2}$$

with $\cdot$ standing for the scalar product. Above, $\mathbf{x_c}$ is the camera position and $\mathbf{e_u}$, $\mathbf{e_v}$, and $\mathbf{e_f}$ are three orthonormal vectors that are defined by the camera angles (azimuth $\phi$, roll $\sigma$, and tilt $\tau$). The camera position and the camera angles conform the *extrinsic* parameters.

In order to account for the lens distortion, instead of the values in expression (1), where subscript "$U$" stands for *undistorted*, the actual *distorted* pixel coordinates are

$$c = \frac{u_{U\star}\left(1 + k_{1\star}d_{U\star}^2 + k_{2\star}d_{U\star}^4\right) + p_{2\star}\left(d_{U\star}^2 + 2u_{U\star}^2\right) + 2p_{1\star}u_{U\star}v_{U\star}}{s_{c\star}} + o_c, \tag{3a}$$

$$r = \frac{v_{U\star}\left(1 + k_{1\star}d_{U\star}^2 + k_{2\star}d_{U\star}^4\right) + p_{1\star}\left(d_{U\star}^2 + 2v_{U\star}^2\right) + 2p_{2\star}u_{U\star}v_{U\star}}{s_{r\star}} + o_r, \tag{3b}$$

where $k_{1\star}$, $k_{2\star}$, $p_{1\star}$, and $p_{2\star}$ are dimensionless parameters for the lens distortion (radial and tangential), and $d_{U\star}^2 = u_{U\star}^2 + v_{U\star}^2$. The relationship between distorted pixels, in Equation (3), and undistorted ones, in Equation (1), comes from their definitions and involves the *intrinsic* parameters $s_{c\star}$, $s_{r\star}$, $o_c$, $o_r$, $k_{1\star}$, $k_{2\star}$, $p_{1\star}$ and $p_{2\star}$. The above Equations (2) and (3) allow for computing the pixel coordinates $(c, r)$ for a given point $(x, y, z)$ if the intrinsic and extrinsic parameters are known. They also allow the reverse mapping to obtain $(x, y)$ from $(c, r, z)$, here with $z$ set to wave-averaged sea level, $z_{\mathrm{msl}}$.

### 2.3.2. Intrinsic Calibration

Each video recorded from the drone, of around 10 minutes, was obtained with smooth motions and a constant focal length, thus the extrinsic parameters had small variations in time while the intrinsic parameters could be assumed constant. In order to calibrate all the frames of a video, it was necessary to obtain their common set of eight intrinsic parameters and, for each frame, its six extrinsic parameters. For this purpose, it was considered a set of $n_I$ images (a "basis" of images) uniformly distributed along the video. These $n_I$ images were first calibrated (i.e., the intrinsic and extrinsic parameters estimated) while using ULISES [21], which minimizes the reprojection error and enforces that all images share the same intrinsic parameters.

### 2.3.3. Extrinsic Calibration

Once the intrinsic parameters were established for each video, the extrinsic parameters were automatically retrieved for each frame based on GCP tracking. Concretely, given a frame $A$ from the video and a frame $B$ from the basis, the first step consisted in estimating an homography that maps the undistorted pixels from $B$ to $A$ [38]. Given this mapping between undistorted pixels, the intrinsic parameters can be used to estimate the transformation between distorted pixels of $B$ into $A$. An homography between two images can be defined if one of these conditions is satisfied: (1) the camera (between the two acquisitions) rotates around its center of projection, with no translation; or, (2) the pixels correspond to physical points that fall in one single plane. In the case under consideration, both of

the conditions were nearly satisfied and it was, therefore, reasonable to look for such an homography. The homography was first estimated by finding pairs of correspondences between the two images and applying total-least-squares solution of a linear system of equations, which minimizes the reprojection error. The point correspondences were found through the feature matching algorithm ORB [39], a modification of the original SIFT algorithm [40] (see Figure 4). Canny edge algorithms, which were used by [41], are not expected to work adequately in the absence of texture, i.e., in the sand surface. The feature matching was done while using the original distorted images, the pixels then being undistorted to find the homography. Not all pairs of points were employed to obtain the homography: for the pairs to be more uniformly distributed along the image, only the best pairs (according to ORB) of each cell of a 6 × 6 grid were considered (Figure 4, yellow circles).

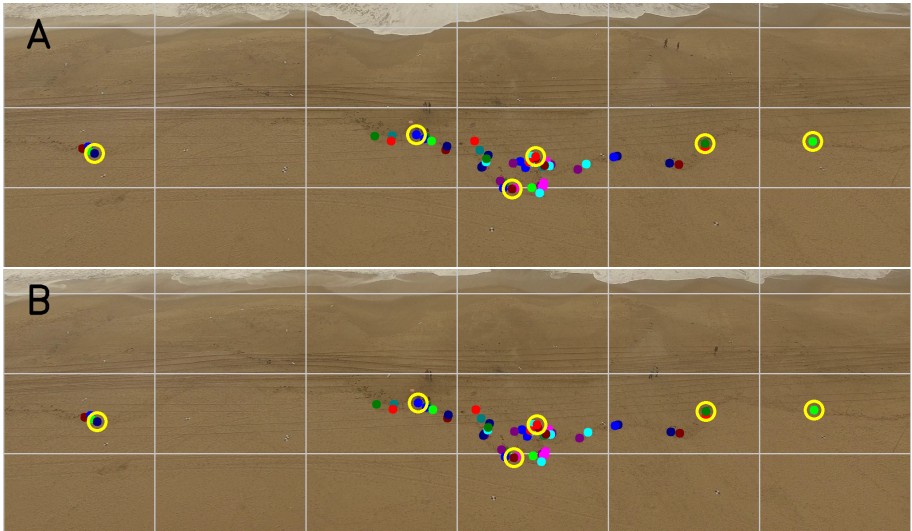

**Figure 4.** Feature matches for an image from the video (**A**) and an image from the basis (**B**) (the images appear cropped for convenience). The matches are plotted with the same color in both images. The 6 × 6 grid is included and the best pair of each cell is highlighted with a yellow circle.

Once the homography from *B* to *A* was estimated (which was not always the case, since it required four pairs at least), the pixel coordinates of the GCPs in frame *B*, known, could be transformed into pixel coordinates in *A*. By visual inspection, the corresponding pixels in frame *A* fell usually on the GCP targets. However, given that the conditions for an homography were not fully satisfied, including a non perfect intrinsic calibration, there were pixels that fell apart from the GCP target in the frame *A*. As an example of this latter situation, Figure 5B shows a zoom of the basis image *B* that is centered in one GCP target and Figure 5A shows the zoom of the image *A* centered at the corresponding pixel through the homography. The GCP target is far from being at the center of the image in Figure 5A, as it should if the homography was correct. To fix this "error", a second feature matching ORB was run between the two zoomed images (Figure 5 shows the pairs of pixels). This comparison allowed for finding the translation to apply to obtain the GCP in Figure 5A. The result is highlighted with a yellow dot, which actually falls on the GCP target in Figure 5A.

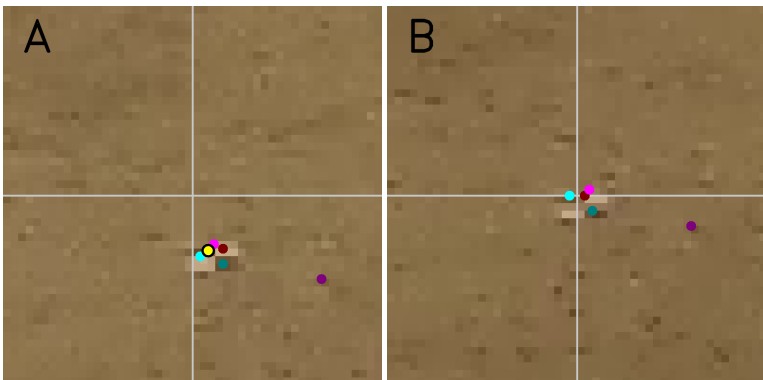

**Figure 5.** GCP tracking after the homography. The center of (**A**) corresponds to the center of (**B**) (where the GCP is) through the homography. The colored dots are the ORB pairs that allow to know the required translation to find the GCP in A (yellow dot).

By comparing an image *A* from the video with one image *B* from the basis, the pixel coordinates were obtained for some of the GCPs (of which the real world coordinates were known). Given that there were $n_I$ images in the basis, where $n_I$ can be greater than 1, different pixel coordinates could be obtained for one single GCP. In that case, the average pixel coordinates were considered for the GCP in *A*. Once the set of a GCPs were found in an image *A*, if the number of GCPs was greater than 6, then the extrinsic parameters were found by minimizing the reprojection error (the intrinsic parameters were already known). This procedure yielded the extrinsic parameters for a (large) subset of images from the video. Because it was not possible to calibrate all video images and the extrinsic parameters showed a noisy behavior, the extrinsic parameters were later interpolated and filtered with a a Butterworth filter, with a characteristic length, $t_f$, of few seconds.

In order to validate the calibration process, 30 images along the video have been calibrated by selecting the GCPs manually and performing the complete calibration process (intrinsic and extrinsic parameters).

### 2.4. Bathymetry Estimation

The uBathy algorithm [17] has been used in order to obtain the bathymetry from the videos. This algorithm departs from the video frames, projected in the *xy*-domain, to obtain the bathymetry following the next steps. First, it decomposes the video into all possible *sub-videos* with a duration $w_t$. For each sub-video: (1) it performs a Principal Component Analysis of the time-wise Hilbert transform of the intensities of the frames in the gray-scale; and, (2) for each of the main modes of the decomposition, and wherever it is possible, it retrieves the wave angular frequency, $\omega$, and the wavenumber, $k$. Once it has run over all the sub-videos, the result is, for each point of the *xy*-domain, a set of $N$ pairs $(\omega_i, k_i)$ from which to infer a value of the water depth $h$. Given the dispersion relationship

$$\omega^2 = gk \tanh (kh),$$

with $g$ the gravitational acceleration, given $N$ pairs $(\omega_i, k_i)$, in order to find $h$ the proposed objective function to minimize is

$$\varepsilon(h) = \sqrt{\frac{1}{N} \sum_{i=1}^{N} \left( h - \frac{1}{k_i} \mathrm{atanh} \left( \frac{\omega_i^2}{gk_i} \right) \right)^2},$$

and it follows that the estimate of the water depth that minimizes the objective function is

$$h_{\min} = \frac{1}{N} \sum_{i=1}^{N} \left( \frac{1}{k_i} \mathrm{atanh} \left( \frac{\omega_i^2}{gk_i} \right) \right).$$

The minimum error, $\varepsilon_{\min} = \varepsilon\left(h_{\min}\right)$, will be used to assess the quality of the retrieved value for the water depth, $h_{\min}$. Hereafter, the subscript "min" is avoided when referring to $h_{\min}$ and $\varepsilon_{\min}$.

In order to apply the uBathy algorithm, a prior step is to project all the frames into the plane $z = z_{\mathrm{msl}}$, with $z_{\mathrm{msl}}$ the wave-averaged mean sea level (the influence of the wave height, which is usually neglected, is studied in [42]). In this case, since the drone had some non negligible movement and the extrinsic parameters vary over time, the projection of a constant pixel domain did also change in time. Figure 3 shows the constant pixel domains for both videos. The spatial domain considered for the uBathy analysis is the intersection in the $xy$ plane of all of the projections of the pixel domain (Figure 6). The values for $z_{\mathrm{msl}}$ for each video were obtained from the deployed pressure transducer.

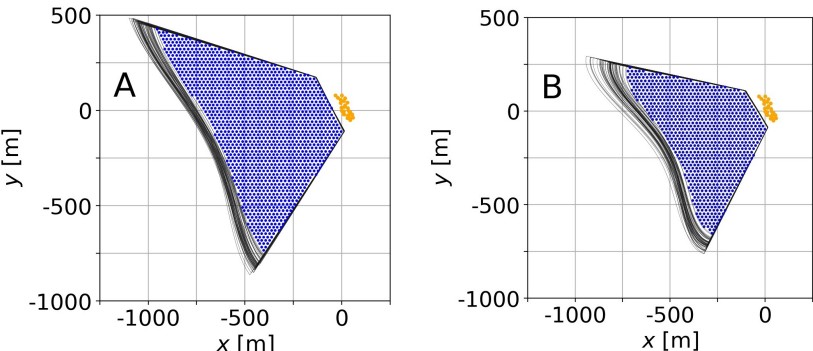

**Figure 6.** $xy$ domain for videos 1 (**A**) and 2 (**B**) as the intersection of all the projections of the pixel domain into the horizontal plane $z = z_{msl}$, mean sea level. Black lines are the boundaries for different video frames, blue dots stand for the mesh within the intersection, and orange points are GCP targets.

A triangular uniform mesh (equilateral triangles with sides of $\Delta = 10$ m) has been considered to discretize the video signal. The domain contains $\sim$5700 points for video 1 and $\sim$3700 points for video 2 (Figure 6 shows the points sparced for clarity). Given that the $xy$-coordinates of the points of the mesh do not correspond, in general, to integer-valued coordinates in pixels, an interpolation in the pixel space has been required in order to obtain the value of the intensity in the gray-scale.

The uBathy algorithm was run for each video splitting the $\sim$10 min in all possible sub-videos with a duration of 90 s (i.e., with $w_t = 90$ s). Further, the time and space radius $R_t$ and $R_x$ to recover, respectively, the wave frequency, $\omega$, and wavenumber, $k$, from the Principal Component Analysis modes were set to $R_t = 2$ s and $R_x = 20$ m following the recommendations put forward by [17]. Finally, the bathymetry at each point is the result of the best fit of the dispersion relationship while using all of the pairs $(\omega_i, k_i)$ obtained in a neighborhood $R'_x = 20$ m of the point for all sub-videos.

## 3. Results

### 3.1. Video Calibration and Stabilization

#### 3.1.1. Intrinsic Calibration

Table 1 shows the values of the intrinsic parameters obtained for both videos using $n_I = 5$ images in the basis. Note that tangential distortion parameters $p_{1\star}$ and $p_{2\star}$ are two orders of magnitude smaller than radial distortion parameters $k_{1\star}$ and $k_{2\star}$, so that, provided the role they play in Equation (3), where $u_\star$, $v_\star$, and $d_\star$ are order 1, and their influence is minor.

**Table 1.** Intrinsic parameters thta were obtained from videos 1 and 2 (with $n_I = 5$) and in laboratory conditions ($L_1$ = indoors; $L_2$ = outdoors and with high obliquity).

|         | $k_{1\star}$ | $k_{2\star}$ | $p_{1\star}$ | $p_{2\star}$ | $s_{c\star}$ | $s_{r\star}$ | $o_c$ | $o_r$ |
|---------|---------|---------|---------|---------|-----------|-----------|---------|--------|
| video 1 | −0.12069 | 0.08505 | 0.00083 | 0.00076 | 0.00085013 | 0.00085472 | 1018.01 | 516.64 |
| video 2 | −0.12184 | 0.09049 | 0.00015 | 0.00077 | 0.00086022 | 0.00087328 | 1015.70 | 531.32 |
| $L_1$ | −0.11178 | 0.06838 | 0.00056 | 0.00550 | 0.00088086 | 0.00088318 | 1018.51 | 538.58 |
| $L_2$ | −0.12940 | 0.08217 | −0.00215 | 0.00551 | 0.00084578 | 0.00084206 | 1017.66 | 539.59 |

Figure 7 shows the results obtained for other values of $n_I \in \{1, 3, 5, 10\}$. This figure also shows the distribution of the intrinsic parameters obtained, for checking purposes, from the 30 aforementioned images of each video. The distributions are, in general, compact, showing that the number and distribution of the GCPs allow for a potentially good calibration of the camera, according to [43]. Besides, the results for the different values of $n_I$ are similar and centered in the distribution (for $n_I = 1$ by chance).

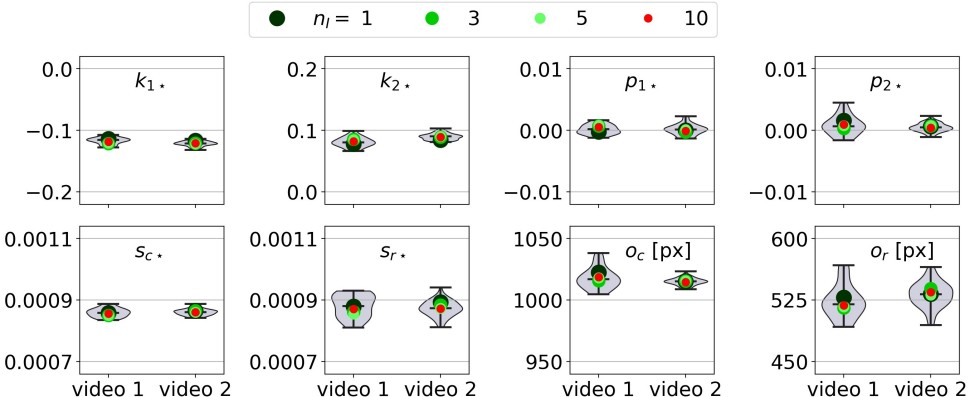

**Figure 7.** Intrinsic parameters for videos 1 video 2 obtained for different values of $n_I \in \{1, 3, 5, 10\}$. The distribution of the intrinsic parameters for other 30 randomly chosen frames is shown in gray.

### 3.1.2. Extrinsic Calibration

The extrinsic parameters ($x_c$, $y_c$, $z_c$, $\phi$, $\sigma$, and $\tau$) obtained following the procedure shown in Section 2.3.3 for $n_I = 5$ are shown in Figure 8 (video 1) and Figure 9 (video 2). For video 1, the procedure yields an extrinsic calibration for 1000 out of the 1225 frames in video 1 (notice a gap near $t \approx 500$ s). For video 2, the procedure gives the results for all 1223 frames and the results are less noisy than those for video 1. Figures 8 and 9 also include the filtered results for $t_f = 5$ s and 10 s.

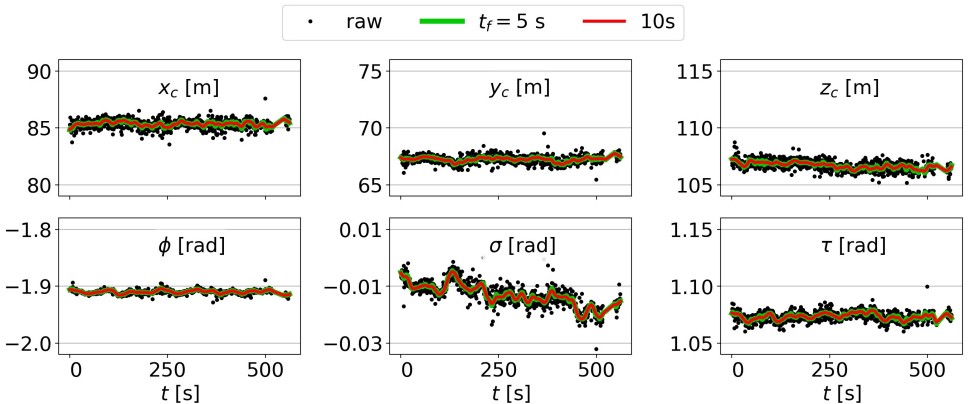

**Figure 8.** Extrinsic parameters obtained for video 1 (black dots), as well as interpolated and filtered results for filtering times of $t_f = 5$ s and $t_f = 10$ s.

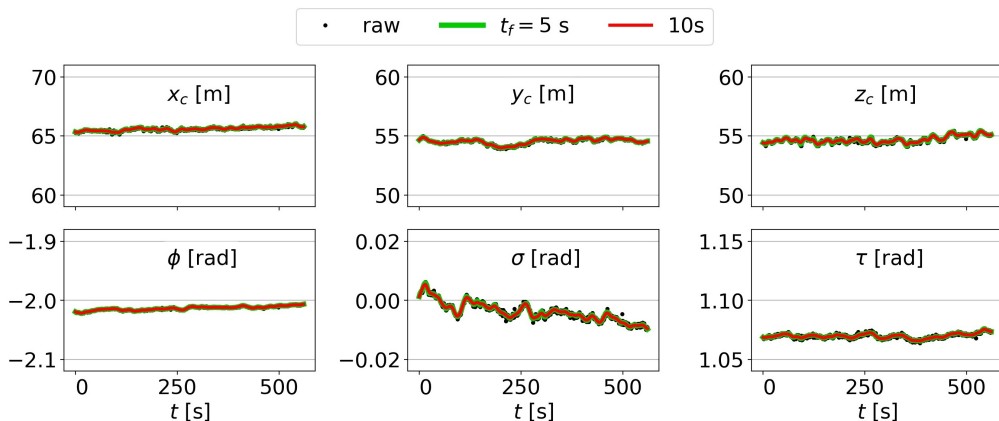

**Figure 9.** Extrinsic parameters obtained for video 2 (black dots), as well as interpolated and filtered results for filtering times of $t_f = 5\,\mathrm{s}$ and $t_f = 10\,\mathrm{s}$.

Once the calibrations are performed, the cross- and along-shore pixel resolution can be computed. These resolutions are the size, or footprint, of a pixel in the cross- and along-shore directions of the $xy$-domain [19,25], and they are of interest to define the mesh on which to infer the bathymetry through uBathy. Figure 10 shows the pixel resolution for the first frame of video 1 and for $n_I = 5$ and $t_f = 5$ s. The largest cross-shore resolution is 5.4 m for video 1 and 6.8 m for video 2. In the along-shore direction, the largest resolution is 3.8 m for video 1 and 4.8 m for video 2. The triangular mesh, with $\Delta = 10$ m, has been defined while taking these pixel resolutions into account.

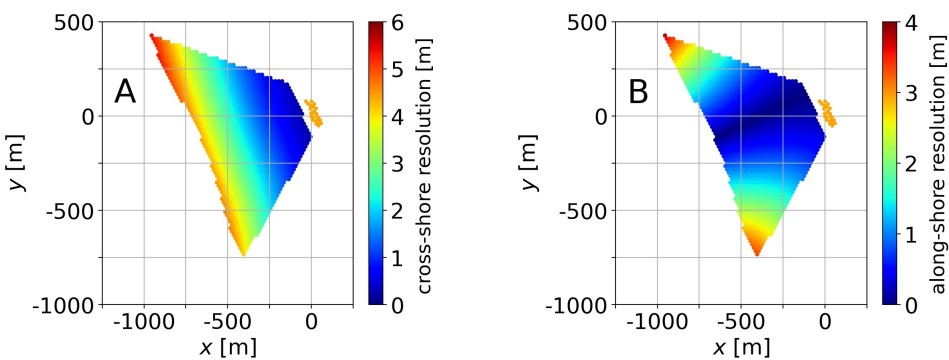

**Figure 10.** Cross-shore (**A**) and along-shore (**B**) resolution for the first frame of video 1 and for $n_I = 5$ and $t_f = 5$ s.

*3.2. Bathymetry Estimation*

Figure 11 shows the bathymetries that were obtained for $n_I = 5$ and $t_f = 5$ s. The main result is shown in the top panels (Figure 11A,B for videos 1 and 2, respectively), which include the bathymetry that was obtained through uBathy as well as the ground truth bathymetry at the path (superimposed). The results from uBathy include some gaps where the algorithm was unable to provide a bathymetry. This is because the signal is not good enough, due to wave-breaking, sun glare, or even stabilization issues. These gaps represent $\sim 11\%$ and 9% of the original domain, respectively, for videos 1 and 2. The top panels also include black zones that correspond to the areas where $\varepsilon > \varepsilon_0 = 2$ m (in Figure 11C,D) and the retrieved bathymetry is disregarded. The reasons why the error $\varepsilon$ is large are the same as mentioned above. Overall, the bathymetry is retrieved, with $\varepsilon \leqslant \varepsilon_0$, for 49% (video 1) and 73% (video 2) of the initial domains. Finally, the bottom panels that are depicted in Figure 11E,F include a scatter plot with the comparison of the in situ bathymetry that was obtained along the path, $h_m$, and the retrieved bathymetry interpolated at the same points, $h_c$ (only if $\varepsilon \leqslant \varepsilon_0$ at that point). The Root Mean Square Error (RMSE) and the bias are

included (a positive bias implies $h_m$ bigger than $h_c$ on average). Table 2 shows the RMSE and bias obtained for all combinations of $n_I \in \{1, 3, 5, 10\}$ and $t_f \in \{0\,\text{s}, 5\,\text{s}, 10\,\text{s}\}$, again for $\varepsilon \leqslant \varepsilon_0$. The percentage of the area of the domain for which $h$ is retrieved with $\varepsilon \leqslant \varepsilon_0$ range from 49% to 66% for video 1 and from 73% to 77% for video 2.

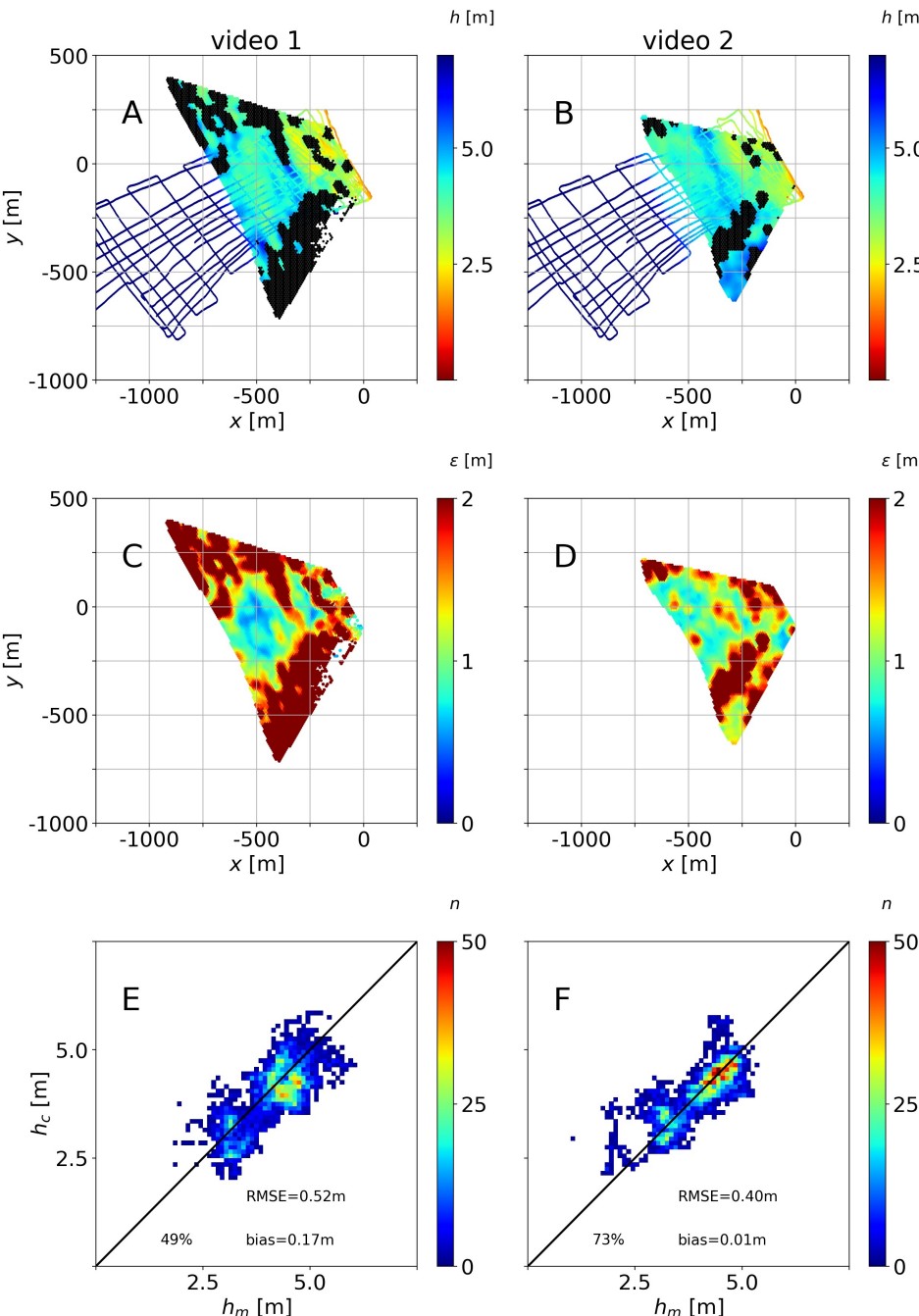

**Figure 11.** Bathymetry estimation for $n_I = 5$ and $t_f = 5$ s for videos 1 (**A,C,E**) and 2 (**B,D,F**): retrieved water depth and ground truth water depth for the boat path (**A,B**), errors $\varepsilon$ (**C,D**)—points with $\varepsilon > \varepsilon_0 = 2$ m are black-masked in plots A and B; measured ground truth, $h_m$, and computed, $h_c$ water depths for the boat path for $\varepsilon \leqslant \varepsilon_0$ (E and F, where $n$ stands for the number of observations at each bin).

**Table 2.** Influence of the number of basis images, $n_I$, and the filtering characteristic time $t_f$, on Root Mean Square Error (RMSE) and bias of the retrieved bathymetry where $\varepsilon \leqslant \varepsilon_0$.

|  | $n_I$ | 0 s | | 5 s | | 10 s | |
|---|---|---|---|---|---|---|---|
|  |  | RMSE | Bias | RMSE | Bias | RMSE | Bias |
| video 1 | 1 | — | — | 0.732 | +0.475 | 0.591 | +0.202 |
|  | 3 | — | — | 0.569 | +0.298 | 0.584 | +0.234 |
|  | 5 | — | — | 0.520 | +0.171 | 0.514 | +0.074 |
|  | 10 | — | — | 0.491 | +0.097 | 0.468 | +0.048 |
| video 2 | 1 | 0.439 | −0.135 | 0.455 | −0.160 | 0.420 | −0.102 |
|  | 3 | 0.407 | −0.065 | 0.384 | −0.008 | 0.389 | −0.049 |
|  | 5 | 0.375 | +0.014 | 0.402 | +0.009 | 0.421 | +0.020 |
|  | 10 | 0.378 | +0.021 | 0.394 | +0.012 | 0.430 | +0.015 |

### 3.3. Intrinsic Calibration: Laboratory Calibrations

Above, the intrinsic calibration has been obtained from a set of $n_I$ frames from the video. However, it is often the case that the intrinsic calibration is provided from "laboratory", while using a set of images that were taken to a chessboard pattern. Here, two other sets of intrinsic calibration parameters are considered, both for images taken with the same camera of the flight, but in dates that are far apart from the flight.

One calibration, which will be referred to as "$L_1$" hereinafter, was performed using images, such as those in Figure 12A, obtained indoors. The second, "$L_2$", considers images that were obtained outdoors and resembling the obliquity conditions of the video (Figure 12B). The (default) intrinsic calibrations obtained from the flight with $n_I = 5$ basis images, as in Table 1, will be denoted as "$F$". The results in this section are presented for $t_f = 5$ s.

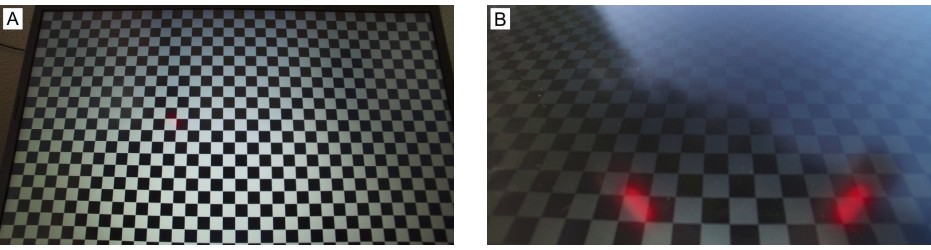

**Figure 12.** hlImages for intrinsic calibrations $L_1$ (**A**) and $L_2$ (**B**).

Table 1 reports the intrinsic parameters obtained. The results of both experiments are similar to each other, and they are also similar to those for $F$. However, there are differences whose impact on the rectification and bathymetry estimation are to be analyzed.

Figure 13 shows the $h_m - h_c$ (measured-computed) histograms that were obtained using the laboratory intrinsic calibrations. Despite the apparently small differences in the intrinsic parameters in Table 1, the results in the inferred bathymetry are different: the intrinsic calibration $L_1$ provides larger values of $h_c$ than real (particularly in the deeper region, where $h_c$ is $\approx$40% larger than $h_m$), while $L_2$ tends to give smaller values (up to $\approx$−30% in the deeper region). The calculated $h_c$ has overall more datapoints below the error threshold $\varepsilon_0$ for $L_2$ ($\sim$70% of the original domain) than for $F$ ($\sim$60%) or $L_1$ ($\sim$30%).

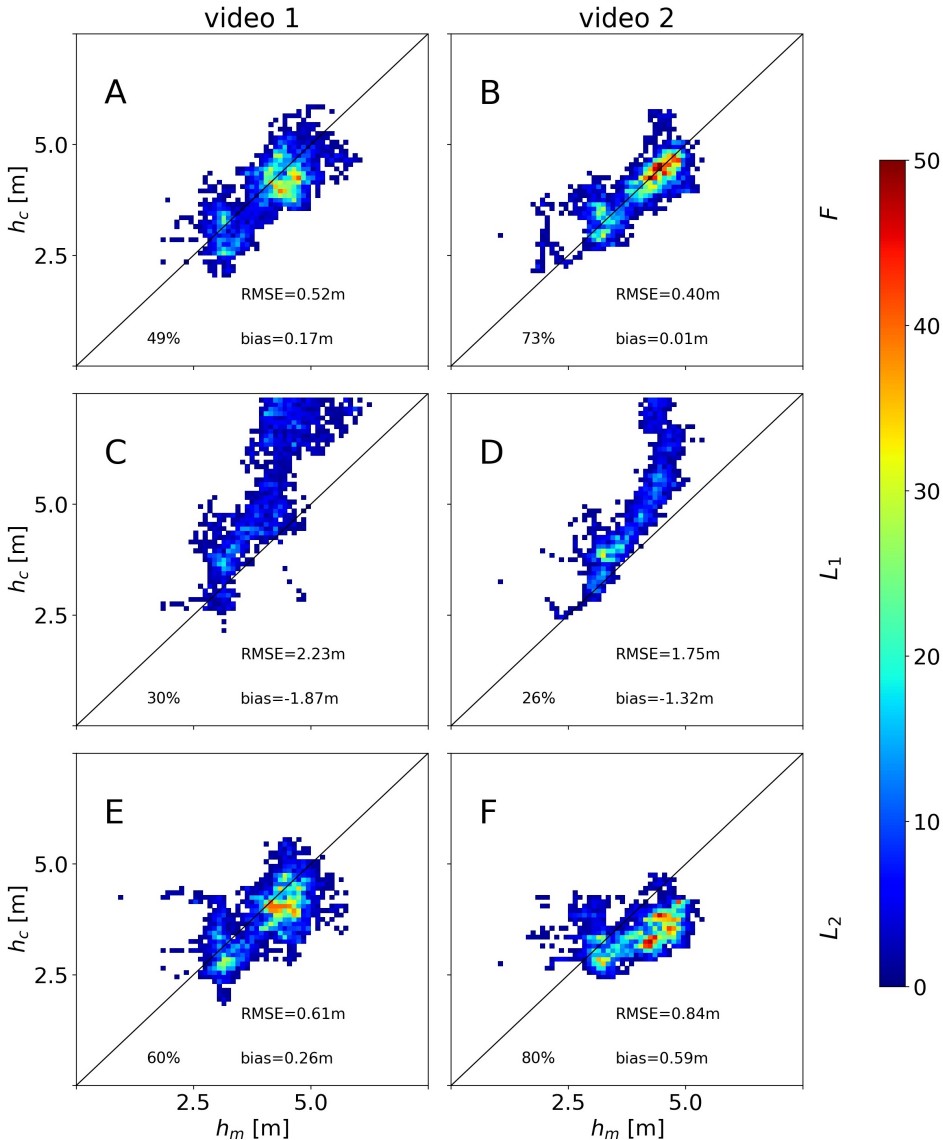

**Figure 13.** Scatter plot of ground truth ($h_m$) and computed ($h_c$) water depths for different sets of intrinsic parameters for $\varepsilon \leqslant \varepsilon_0$. Results for video 1 (**A,C,E**) and video 2 (**B,D,F**) and for intrinsic calibrations $F$ (**A,B**), $L_1$ (**C,D**) and $L_2$ (**E,F**). The plots include the percentage of points of the original domain that are retrieved, as well as the RMSE and bias.

In order to understand where the above differences come from, all of the pixels of each video frame were projected into the plane $z = z_{\mathrm{msl}}$ both via calibration $F$ and calibration $L_i$, and the distances (in the $xy$ space) between both projections were then computed. Figure 14 shows the decimal logarithm, $\log_{10}$, of these distances as expressed in meters for the first basis image of both videos. Because the calibration procedure uses the GCPs, the distances between both projections are very small at the region around the GCPs. However, in the region of interest (boxes in the plots), the distances can reach values of tens or even hundreds of meters. The higher is this distance, the more potential discrepancies between $F$ and $L_i$. According to Figure 14, for $L_1$, some pixels can fall, when projected to $xy$, hundreds of meters apart of their position when projected while using $F$, which explains the particularly bad behavior of $L_1$ depicted in Figure 13 .

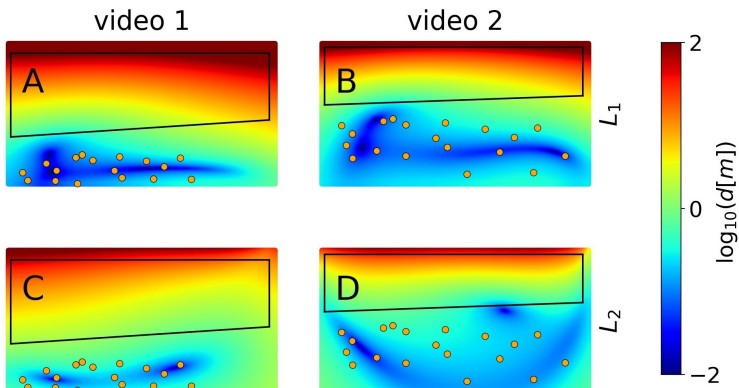

**Figure 14.** Decimal logarithm of the *xy*-distances, *d* (in meters), of the projected pixels using $F$ and $L_i$ for video 1 (**A**,**C**) and video 2 (**B**,**D**). The plots also include the pixel-domains for each video (black boxes) and the GCPs (orange points).

There is an alternative way to understand the results shown in Figure 13, which will also explain that the errors for $L_1$ are in excess. Once $L_i$ is considered, and the calibration of the extrinsic parameters has been carried out for the video, as described in the methodology, the *xy* domains can be obtained (as in Figure 6). Figure 15A,B show the domains that were obtained for $F$, $L_1$, and $L_2$ for both videos. The domains for $L_2$ and $F$ are similar, whereas the domain for $L_1$ is ≈20% larger. Hence, while using $L_1$, the wavelength will appear longer (besides in a different *xy*-position) and the inferred water depth will be larger. Otherwise, for $L_2$ in video 2, the domain is smaller, and the wavelengths (and inferred water depths) will also be smaller. Hence, Figure 15A,B explain the behavior depicted in Figure 13. Note that, while two different intrinsic calibrations that provide the same domain do not necessarily have similar calibrations (there can be, although unlikely, inner compensations), two very different domains do actually imply that the calibrations behave differently.

### 3.4. Camera Governing Equations

The influence of the governing Equation (3), which includes the intrinsic parameters, has also been analyzed. When dealing with video monitoring images, where the GCPs are often scarce and not well distributed along the image, some reasonable simplifications of the model have shown to work more robustly for some cameras [43]. Three different simplifications have been considered here:

- $S_1$: squared pixels (i.e., $s_{c\star} = s_{r\star}$);
- $S_2$: $S_1$ + no decentering (i.e., $o_c$ and $o_r$ at the center of the image); and,
- $S_3$: $S_1$ + $S_2$ + parabolic radial distortion only (i.e., $k_{2\star} = p_{1\star} = p_{2\star} = 0$).

The *xy*-domains that were obtained for the original model ($F$) as well as for the three simplifications are shown in Figure 15C,D. In all cases, the intrinsic parameters are obtained from the flight, with $n_I = 5$ and $t_f = 5$ s. From Figure 15C,D, it can be anticipated that, except for $S_3$, the differences should be small. The results (in Table 3) confirm that the differences are, in general, below 10 cm in RMSE, except for $S_3$. Interestingly, for video 1, worse conditioned in terms of GCPs distribution, simplifications $S_1$ and $S_2$ improve the bias while keeping similar values for RMSE.

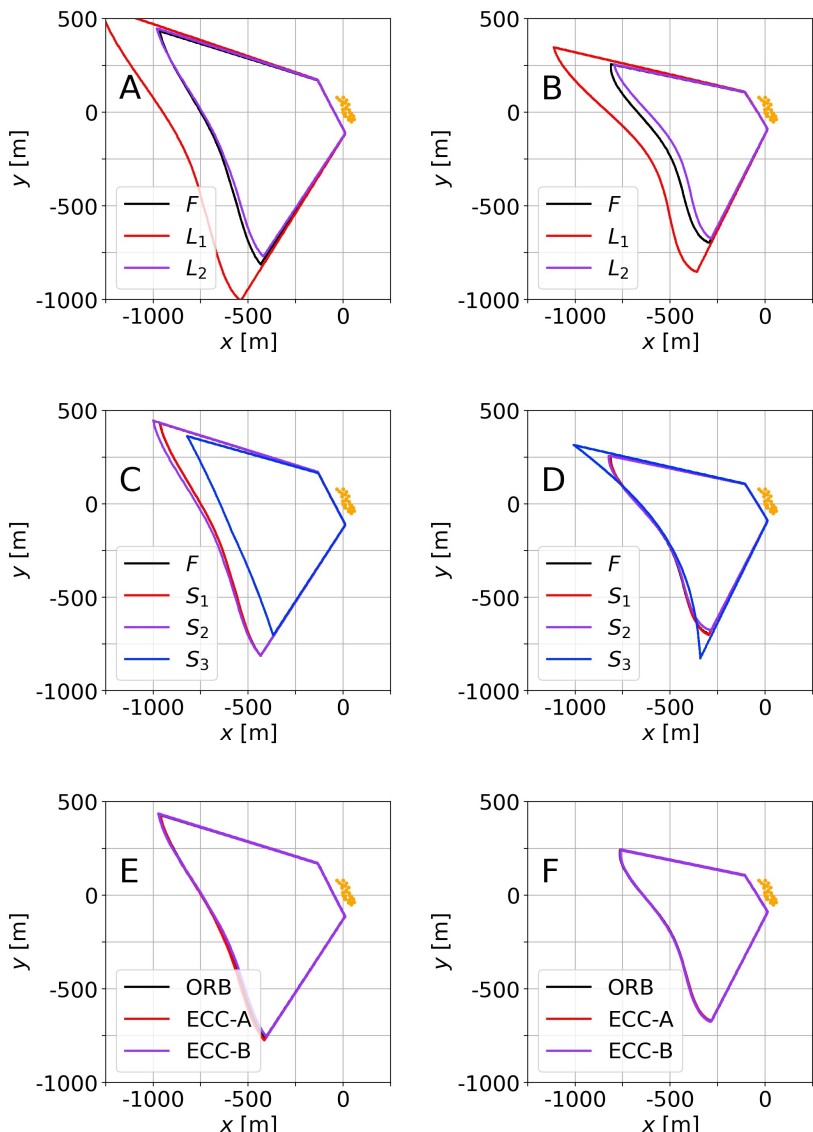

**Figure 15.** Influence of several aspects on the domains obtained for video 1 (**A**,**C**,**E**) and video 2 (**B**,**F**,**D**): influence of using the intrinsic calibration from field (*F*, with $n_I = 5$) and from laboratory ($L_1$ and $L_2$) (plots **A**,**B**); influence of the simplifications of the camera governing equations (**C**,**D**); and, influence of the GCP tracking approach (E and F). In all cases, GCPs are marked with orange points.

**Table 3.** RMSE and bias, for $\varepsilon \leqslant \varepsilon_0$, of the inferred bathymetry for different simplifications in the governing equations.

| | Simplification | RMSE [m] | Bias [m] |
|---|---|---|---|
| | $F$ | 0.520 | +0.171 |
| video 1 | $S_1$ | 0.521 | +0.106 |
| | $S_2$ | 0.529 | −0.040 |
| | $S_3$ | 1.304 | +1.309 |
| | $F$ | 0.402 | +0.009 |
| video 2 | $S_1$ | 0.410 | +0.108 |
| | $S_2$ | 0.439 | +0.072 |
| | $S_3$ | 0.511 | +0.236 |

### 3.5. Extrinsic Calibration: Homographies and GCP Tracking

Finally, the tracking of the GCPs has also been performed in alternative ways. The homography between two (undistorted) images has been obtained while using the Enhanced Correlation Coefficient (ECC) image alignment [44] as an alternative to ORB. This method (ECC) is recommended in images where the textures are not clear. For each image of the video (previously undistorted), ECC was performed with each of the $n_I$ undistorted images of the basis. All cases where the resulting correlation coefficient ($cc$) was below 0.5 were disregarded. Furthermore, two different options for working each frame of the video are considered: in the first option, "ECC-A", all of the images of the basis that give $cc \geqslant 0.5$ are considered to track the GCPs. In the second one, "ECC-B", only the image giving the highest is used (as long as $cc \geqslant 0.5$). Once the homographies are obtained through ECC, it follows the same procedure as for ORB above (illustrated in Figure 5). For ECC-A, if one GCP can be tracked through several images of the basis, then the averaged position was considered. The resulting $xy$-domains for all three methods are indistinguishable for video 2 (Figure 15F), and nearly indistinguishable for video 1 (Figure 15E), so that the results should be similar in terms of the obtained bathymetry. Table 4 shows the RMSE and bias. Except for the bias in video 1, which is notably improved while using ECC, the results are very similar to those for ORB. Regarding the bias for video 1 using ORB, Table 2 already indicated that, for this video, it was convenient to increase $n_I$ and $t_f$ (here $n_I = 5$ and $t_f = 5$ s). The computational cost to obtain the homographies, for our images, ECC was one order of magnitude more expensive than ORB.

**Table 4.** RMSE and bias of the inferred bathymetry for different GCP-tracking methods.

|  | Method | With Refinement | | Without Refinement | |
|---|---|---|---|---|---|
|  |  | RMSE [m] | Bias [m] | RMSE [m] | Bias [m] |
| video 1 | ORB | 0.520 | +0.171 | 0.944 | +0.005 |
|  | ECC-A | 0.529 | +0.053 | 0.714 | +0.018 |
|  | ECC-B | 0.520 | +0.025 | 0.558 | +0.006 |
| video 2 | ORB | 0.402 | +0.009 | 0.481 | −0.091 |
|  | ECC-A | 0.448 | +0.015 | 0.400 | −0.065 |
|  | ECC-B | 0.397 | −0.007 | 0.434 | −0.047 |

## 4. Discussion

The results presented in the previous section show that the intrinsic camera calibrations providing the best results are those that are obtained while using video images from the flights, instead of those resulting from the conventional use of chessboards in the laboratory. This is a fact that may have been unexpected at first sight. However, the characteristics of the camera depend on the environmental conditions (temperature, humidity, or atmospheric pressure), and they will generally be different in the laboratories than during the flights. As a practical consequence, videos from cameras that have not been calibrated in laboratory conditions and for which the characteristics are not known, can be used. It is sufficient and, in fact, more convenient (at least for a configuration like the one here, where the GCPs are grouped in a part of the image), in order to use the georeferenced points (GCP) that are, in any case, necessary for the extrinsic calibration process.

The distribution of the GCPs in the images is a crucial factor in obtaining good calibrations, and is the reason why the results of the video 2 are actually better than those of the video 1. It is recommended to prepare the field campaigns to include a large portion of GCPs in the images. It is not very important the number of GCPs to be large, but it is essential to distribute them widely. Under these working practices, it is enough to manually calibrate between three and five images to achieve a high quality intrinsic calibration (see Figure 7). The calibration model of the camera can be simplified (Table 3) with square pixels and no decentering, but quartic distortions are necessary. Models for image calibration that only include parabolic radial distortion should be avoided.

In the methodology presented to identify GCPs, the different methods for performing the homographies give similar results. The possibility of using the GCPs directly from the homography, i.e., avoiding the refinement procedure illustrated in Figure 5, has also been considered (right hand side Table 4). The results are, in general, better when the refinement is applied, particularly in video 1 (which is worse conditioned), except for ECC-B. Therefore, performing a refinement to locate the GCPs at the images is recommended. In regard to using ORB or ECC (-A or -B), the results are not conclusive, so that ORB is recommended if the computational time is an issue. In any case, the influence of the intrinsic calibration shown in Section 3.3 is much more important than the tracking algorithm.

The subsequent smoothing and temporal filtering of the extrinsic calibrations improves the bathymetry estimation. For the video 2, the improvements are not very significant. However, for the video 1, it is essential. The underlying reason is the location of the GCPs that are affecting both the intrinsic and extrinsic calibrations. For video 1, which is worse conditioned (cornered GCPs and more noisy extrinsic calibration in Figure 8), increasing the number $n_I$ of basis images improves the results, and applying some time filtering is necessary (there are no results for $t_f = 0$ s). For video 2, where the GCPs occupy nearly half of the image and the raw extrinsic calibration shows a smooth behavior (see Figure 9, for $n_I = 5$), bathymetries with RMSE $\approx 40$ cm and bias $\approx 1$ cm are obtained for $n_I \in \{3, 5\}$ and for $t_f = 0$ s and 5 s. Recall that uBathy, which performs a Principal Component Analysis decomposition, already performs some time filtering of the signal.

The videos have been processed by interpolating the pixels into a horizontal triangular grid of 10 m with equally spaced points throughout the domain. On this same grid, the bathymetry has been inferred and, therefore, the estimated bathymetry also has a spatial resolution of 10 m. This distance is subject to the footprint of the pixels, where in the outer region is up to 7 m, and to the wave length of the waves, which in the domain ranges from a few tens of meters in the shallowest area to hundreds of meters in the outer area. The water depths ranged from 2 m to 6 m. UBathy calculates the depth from the linear wave dispersion relation, so that we assume gentle variations over of the order of the wavelength. Variations in the spacing of the grid could be accounted for in later works, given the variation in the domain of the result from the images and the wavelength. The accuracy of bathymetry estimation is difficult to evaluate. In this work, the errors in the bathymetry have been used in order to evaluate the accuracy of the calibration, the goal of the paper, through the projection of the images over the water level. Therefore, it is not possible to isolate the accuracy of the calibration from that of the bathymetry. However, the RMSE and bias are generally lower than those of other studies (e.g., [30–32]) (with RMSE in the range of 0.25 to 0.50 m and biases from 0.2 to 0.3 m).

Finally, the applicability of the methods that are presented in this study should be assessed. The method has been developed for videos for which the intrinsic calibration of the lens was unknown and for conditions in which the orientation and position of the camera (extrinsic calibration) were unknown as well as variables of time. Therefore, the same method can be applied to videos that are obtained without a rigid base (e.g., mobile phones) or for fixed cameras (public webcams or "Argus" stations). For the second case, smoothing and averaging of the extrinsic calibrations is certainly unnecessary. It is crucial that a significant proportion of the image contains GCPs in order to obtain a final realistic bathymetry, as mentioned above. In the case of fixed stations, where there are often GCPs in fixed positions permanently, the method that is based on homographies and the correction of the position of the GCPs can be used to automatically recalibrate the cameras and to correct daily and seasonal variations due to changes in ambient conditions.

## 5. Conclusions

In this paper, video images from UAVs have been processed in order to obtain coastal bathymetries. The effect of GCPs distribution in the field, camera model, and intrinsic and extrinsic calibrations on bathymetric estimation is analyzed. From the calibrated videos, the bathymetry has been obtained while using uBathy. The bathymetries inferred from the

video have been compared with a bathymetric survey made during the video acquisition. A proper calibration of the videos allows for obtaining accurate bathymetries with root mean squared errors that are below 40 cm with biases of a few centimeters.

Intrinsic calibration is the most critical element of the process. The optimal intrinsic camera calibration has been obtained from manual calibration of a few frames of the videos (three or five) with the GCPs located on the dry beach. Intrinsic calibrations from chessboards are of lower quality and they do not allow for reliable bathymetries. With regard to the camera model, it is crucial to include quadratic and quartic distortions (degree 2 and 4 of the distance to the camera center). The extrinsic calibration of each image has been performed by an automatic detection of the GCPs. The localization of the GCPs has been executed in two steps. First, an homography between the manually calibrated images and each image of the video and, second, a correction of the estimated location from the homography to the actual position. The precise procedure for finding the homographies is not crucial, although the computation times are reduced by using ORB as compared to ECC, since the aim is to obtain an estimate for the location of the GCPs. Conversely, a further refinement of the point location is critical. A subsequent time filtering, order of 5 s, of the extrinsic calibration from each image allows for recovering and stabilizing the positions and orientations of the UAV during the flight and, finally, the correct projection of the images in the real world. To summarize, in all cases considered here, and for both flight conditions, the default calibration (*F*) produces robust results and provides better bathymetries than those that are obtained from laboratory intrinsic calibrations.

**Author Contributions:** The first two authors have equally contributed to the work. Conceptualization, G.S., D.C. and T.A.P.; methodology, G.S., D.C., T.A.P. and F.M.-N.; software, G.S., D.C. and F.M.-N.; validation, G.S., D.C. and I.G.-L.; formal analysis, G.S., D.C. and F.M.-N.; investigation, G.S., D.C., T.A.P. and F.M.-N.; resources, G.S. and D.C.; writing—original draft preparation, G.S., D.C. and T.A.P.; writing—review and editing, G.S., D.C., T.A.P., F.M.-N., I.G.-L., J.M. and R.D.; visualization, G.S., D.C., T.A.P. and R.D.; supervision, G.S., D.C., T.A.P., F.M.-N., I.G.-L., J.M. and R.D.; project administration, G.S. and D.C.; funding acquisition, G.S., D.C. and T.A.P. All authors have read and agreed to the published version of the manuscript.

**Funding:** This research was funded by the Spanish Government (MINECO/MICINN/FEDER) grant numbers RTI2018-093941-B-C32, and RTI2018-093941-B-C33, PID2019-109143RB-I00 and FEDER-UCA18-107062. G.S. and R.D. belong to CRG on Littoral and Oceanic Processes, supported by Grant 2017 SGR 863 of the Generalitat de Catalunya. Part of this work and I.G.-L. was supported by a grant funded by the European Commission under the Erasmus Mundus Joint Master Degree Programme in Water and Coastal Management (WACOMA; Project num. 586596-EPP-1-2017-1-IT-EPPKA1-JMD-MOB).

**Institutional Review Board Statement:** Not applicable.

**Informed Consent Statement:** Not applicable.

**Data Availability Statement:** All data are available under request to the first author.

**Acknowledgments:** P. Zarandona, M. Aranda, J. Benavente and L. Barbero are acknowledged for their help during the field campaign. Logistic support was provided by the Drone Service of the Univeristy of Cádiz.

**Conflicts of Interest:** The authors declare no conflict of interest.

## Abbreviations

The following abbreviations are used in this manuscript:

| | |
|---|---|
| GCP | Ground Control Point |
| NAO | North Atlantic Oscillations |
| RMSE | Root Mean Square Error |
| RTK-GPS | Real-Time Kinematic Global Positioning System |
| UAV | Unmaned Aerial Vehicle |

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
