# Peer review of "The Influence of Camera Calibration on Nearshore Bathymetry Estimation from UAV Videos"

_remotesensing, doi:10.3390/rs13010150_

Round 1

Reviewer 1 Report

This paper investigates the effect of camera calibration on bathymetric inversion using optical imagery obtained with a UAV. The paper demonstrates that a proper calibration routine can improve accuracies of inverted depths collected by a UAV. This work has potential and is certainly of interest for the readers of this journal.  However, major revisions are needed before publishing. 

First, the paper is not well-structured. If calibration and stabilisation is the focus of this paper, this should be well-introduced in the introduction, including several references (instead of only one). What is calibration, how is this done, what different approaches exist? etc. I imagine that the basics of calibration could actually be moved from the methodology (section 2.3) to the introduction, although it would need rewriting as this part is now difficult to understand. Results section 3.1 would fit better in the methodology, as it is a description of the ground truth data set. The discussion is a sensitivity analysis on the calibration results. One could argue that these are results. Anyway, a proper discussion section putting this work in perspective of existing literature is clearly missing (For example, how do bathymetric accuracies compare with other UAV-obtained bathymetries? How are different calibration routines needed for UAVs than for land-based or satellite stations?) 

Second, the work is not well explained and, in this form, cannot be understood by the reader. A thorough check on the language, typos, inconsistencies in the text should be done by the authors and co-authors before submitting the revised paper. The current version contains too many textual errors and incorrect phrases.

—————————General comments ———————

A validation of computed water depths with measured water depths is only shown for -2.5 m to -5 m (Figure 10E,10F), although in section 3.1 is mentioned that depths until -7.5 m are considered. I suppose that depths of -5 to -7.5 have errors e>e0 and are therefore not shown. Actually it would be interesting to see how these larger depths relate to measured depths and whether they are generally too large or too small. This would be valuable information. In addition, depths can be validated across a wider range than -2.5..-7.5m if dropping the restriction of only considering ‘intersects’ of in situ depths. I think it is valuable to show an approximate ‘inner error’ of in situ depths in the discussion, but I do not see the value of only analysing depths whereof the inner error is known. I think it is too restrictive. A validation for such a limited range is a weak point of the paper. 

A comparison is shown between two videos, but why the results are different for the two videos is not clearly discussed. Some explanations at different points in the text are given. The paper would benefit by a separate subsection in the discussion on this topic. Such a section could start with 1-2 summarizing phrases about the main differences in bathymetric accuracies for video 1 and video 2, and could then discuss possible causes (effect of GCP distribution, height of UAV, stability of UAV x,y,z position, weather conditions). This would be valuable information for other studies. 

Language is too informal at times. Please avoid language such as ‘very’, ‘much’, ’whole’ etc. Here are a few examples: Much less noisy (line 224), very good (line 245), whole (line 251), very (line 272), much larger (line 272), very similar (line 309). This list is incomplete, please check throughout the paper.

Language is indirect. Try to write with less words and to be more clear at the same time. Often phrasing like ‘in regard’, ‘’in terms of’ is not necessary, and  ‘in order to’ can be simply written as ‘to’. For example: Lines 273-275: In regard the percentage of points of the original domain-…- with ….gives significantly lower values —> Calculated hc has overall more datapoints below the error threshold e0 for L2 (…%) than for F (..%) or L1 (..%)

Use of -…..- (e.g. lines 239, 241, 284, 291…). I find the dashes confusing, especially because they are used in combination with several commas and/or brackets. I recommend using commas, combined with round brackets where necessary. 

ABSTRACT

The abstract should contain the goal of the study. This should be the same goal as mentioned in the introduction. 

INTRODUCTION

The introduction could be better structured. 

2nd paragraph: After making the point that in situ techniques can be used to obtain accurate bathymetries, but against high cost…, the next phrase should state that remote sensing techniques can be used as alternative as they can collect image data of the sea surface and beach whereof bathymetry can be retrieved against lower cost and under a wider range of weather conditions (In other words, clarify how remote sensing can be an advantage over in situ). Then, mention the different types of imagery that has been used for bathymetric retrieval (lidar, x-band radar, optical, satellite (is that multispectral? If satellite references are based on optical or X-band radar data, there is no need to mention satellite. Just include the references with optical or x-band). Then conclude the paragraph with line 22-24 (‘The preference …. and the budget’).

3rd paragraph: describe the different platforms used to collect optical imagery (i.e. land-based stations, satellites and UAVs)

4rd paragraph: bathymetric retrieval (lines 30-36)

5th paragraph: challenges on camera calibration (line 41-48)

6th paragraph: lines 49-58 

METHODOLOGY

As a general rule the methodology should be written in past tense, without the use of ‘we’ and ‘us’.

Section 2.4 bathymetry interference. First the uBathy algorithm should be described in brief. Then settings that are different than default or algorithm adjustments can be mentioned. Text about the pixel domains and mesh may fit better in section 2.2 (data collection)

Be consistent with naming. Use video 1 and video 2 throughout the paper and don’t switch to ‘first video’ and ‘second video’

RESULTS

Section 3.1 this section is no result, but belongs in a description of the data set (e.g. section 2.2)

Figure 10. What is n on the color axis?

DISCUSSION

Table 3. Merge table 1 and 3. This would allow better comparison of the intrinsic parameters obtained during flight or in the lab.

Lines 276-278 and Figure 13. I don’t understand. In general, the text in lines 276-285 is not well written and needs revision. For example, a verb is missing in line 282 (the higher… discrepancies). Also, lines 282-285 are unclear and poorly phrased. In line 285, what is error in excess or deficiency? 

Figs 14-16 could be combined in a single figure of 6 subplots. This would directly give insight in the sensitivities to intrinsic calibration, extrinsic calibration and the model. 

Why not showing bathymetric maps for the different sensitivities in several subplots? Possibly showing this for only one of the videos is already insightful.

CONCLUSION

It is important to mention the limitations of the study in the conclusions. For example, ‘quality bathymetries with root mean square errors below 40 cm and small biases of a few centimeters’. Yes this is true for specific depths of -2.5 to -5 meters. Please, mention this. Such a nice rmse was reached because values with high errors (error>2) were removed.

---------------Specific comments------------------

Line 2 Unmaned —> Unmanned

Line 4 procedure —> methodology

Line 4-5 and the results are compared —> and compare the results 

Line 4 ‘results’ is very vague, please be more specific

Line 8 Lens aberration. This term is not mentioned anywhere else in the paper. Please be consistent in terminology/wording.

Line 13 being able to measure —> measuring

Line 13 problem —> challenge

Line 16  I don’t understand what delineation of the active shore face is.

Line 16 shorefare —> shore face?

Line 17 validating —> the validation

Line 19 [1,9, e.g.] —> [e.g., 1,9]

Line 21 I think the reference in line 20-22 should only include bathymetric retrieval. In my opinion, [11] Levoy et al. (2013) should not be mentioned here. 

Line 22 video images —> optical video images

Lien 23 size of the area to be analyzed —> dimensions of the study site

Line 24 the weather conditions —> the predominant weather conditions 

Line 24 budget —> available budget

Line 27 a coastal site —> coastal sites

Line 27 site, covering —> collecting imagery that covers…and has a time sampling…..

Line 28 ‘They..’ Who? Monitoring stations? Images?

Line 28 readily. I think images cannot readily be applied. Always processing is needed to retrieve proxies that can serve to analyse coastal change. I would rephrase to ‘imagery has been used to analyse….’

Line 28-29 shoreline detection and analysis of coastal variability —> shoreline variability

Line 29-30 ‘morphodynamics of beach systems’. Isn’t beach morphology a better description?

Line 30 all methods to —> all methods that 

Line 32 estimating —> the estimation

Line 33 which is constant in the whole domain —> nearly spatially invariant. Space varying —> spatially varying

Line 34-35 ‘analyzed the video…video as a whole’. I don’t understand

Line 37 is making possible —> allows

Line 39 high vantage point. I don’t understand. Does that mean a high point where cameras can be installed? Cameras can be installed on top of a camera tower. So, this is not a valid argument. 

Line 40 punctual survey? Does that mean a single survey?

Line 40 obtained from —> recorded with

Line 41 the bathymetry —> bathymetry

Line 53 is based on ubathy —> is obtained with

Line 61 a 3kmlong —> a 3km long

Line 63-64 average annual significant offshore wave significant height —> average annual offshore significant wave  height. 

Line 64 What is the annual peal period? Introduce symbols for waveheight, peak period and direction in the same sentence.

Line 65-67 this sentence is confusing. remove ‘monthly average’ and rewrite: During the storm season (), Hs is .., Tp is and theta is …. During the calm season (), Hs is.., Tp is, and theta is …

Line 67 Inter annual storm climate —> The interannual wave climate 

Line 68 More storm present —> higher storm frequency?

Line 134 The term homography should be explained. 

Line 174 bathymetry inference —> bathymetry inversion. Use existing terminology. Also elsewhere in the text

Line 175 To apply uBathy —> to apply the uBathy

Line 182 A triangular …10 m). This sentence is incomplete

Line 183 (Figure 5….clarity). Plotting details should appear in the figure caption not in the text. Check this throughout the text.

Line 184-185 For the …corresponding frame. I don’t understand this sentence

Line 253 The sentence ‘All steps.. introduced above’, basically says: reader, we have simplified it for you as much as possible, because you would not be able to understand…?. This sentence is not necessary.

Line 270 The results…purposes. Plotting details should appear in the figure caption not in the text. Please remove.

Line 273 h—>hc?

Line 272 Very different. Much larger. This is qualitative language. Differences and similarities should be quantified

Line 273 Smaller. Again, change into quantitive description

Line 287-289 The grammar is not correct.

Line 290 Could be rewritten: The domains for L2 and F are similar, whereas the domain for L1 is larger.

Line 301-302 The algorithm…of the basis. The grammer is not correct.

Line 304 …to work each frame. I don’t know what that means.

Lien 306 ‘highest cc>0.5’. remove >0.5?

Line 302 ‘basis’. What is basis? Was this earlier called ‘base’?

Line 309 Indistinguishable? Figure 15 shows that the domains are similar for the three different GCP-tracking methods for both video 1 and video 2.

Line 309-310 ‘Table 4 -‘2 steps’-shows the results in t terms of RMSE and bias —> Table 4 shows the RMSE and bias of ….

Line 334 Interestingly,.. values for RMSE. Bias is a measure that can reduce if positive and negative errors cancel out each other. If simply looking into the depth maps of F,S1 and S2, the authors could see why the bias reduces and explain that in the text. 

Line 339-341 The focus…obtained bathymetry. This sentence is not necessary if the next sentence is slightly rewritten into: We analysed the effect of GCP distribution in the field, camera model and intrinsic and extrinsic calibrations on bathymetric inversion.

Line 344. ‘Sea bottom survey’. Please use same terminology as elsewhere in the paper.

Line 354 quality bathymetries —> accurate bathymetries

Reviewer 2 Report

The authors describe an alternative approach for estimating nearshore bathymetry by exploiting the configuration of sea-surface waves recorded on video data. They introduce their approach and the study area in a comprehensive way and their results are also presented sufficiently. However the Discussion section is mainly focused only on the technical aspects of the processing and thus it is lacking a paragraph where the overall quality and broader implications of the results should be discussed. I suggest that the authors summarize their Discussion about the interior/exterior calibration in one paragraph and then they provide another paragraph discussing the following points:

1) what are the horizontal and vertical resolutions of the estimated bathymetry?

2) Which factors are affecting the horizontal resolution and how does it change over the study area?

3) what are the sources of error and how the bathymetric uncertainty can be incorporated in the results?

4) what are the most prominent applications of the described methodology? (e.g. real-time monitoring?). Can this method be used already in the field or is  it at experimental stage?

These are considered as major issues and the authors need to address the above points sufficiently before their paper is further considered for publication.

Reviewer 3 Report

I believe it is a job well done that really uses data to define metrics for evaluating the photogrammetric procedures used to obtain data. it also shows a deep computational capacity, necessary to analyze and manage large amounts of data deriving from remote sensing techniques.

Please verify line 282, it seems to be missing a verb, and please check double words in line 323 and 336.

Finally, I suggest using "focal length" instead of "zoom" in line 123.

Reviewer 4 Report

Dear authors,
I congratulate you on a great study that is presented in your paper.
The paper is well written and properly illustrated and referenced.
I do not have any major comments or suggestions. The only thing that I would like to mention is the bathymetry colour scales. Traditionally colour ramp for bathymetry would have red colours for shallower regions and blue for deeper areas. The scheme that is used in the paper can be confusing for a reader. This refers both to figures 6 and 10. Also figure 10 that summarizes the study is shown in such size that does not allow to properly see the results.

Round 2

Reviewer 1 Report

I wish to thank the authors for the rather extensive revisions in the limited time. The authors certainly improved the text, structure and figures in the paper. I consider the paper publishable after a thorough spelling check and a consideration of the following minor comments:

title: a stronger title could be chosen, please reconsider. (e.g. 'the influence of camera calibration on nearshore bathymetry estimation from UAV videos') 
abstract: A phrase is missing that introduces the importance of camera calibration and stabilisation (this is what the paper is about). A phrase could be added about the findings, and potentially, implications. 
line 61-63: 'experimental data includes an echo-sounder bathymetry performed'
line 68: 'Section 3 also4 includes'
line 209: 'os'
line 301: 'basise'
line 308: 'Tables and 1'
line 426: could this statement be illustrated by mentioning the numbers of bias and RMSE of the references [30-32]?
line 461-463: this sentence is a bit too vague to be a concluding sentence with 'all cases' and 'other calibration methods'. 

Reviewer 2 Report

The authors have addressed the requested comments sufficiently and they have incorporated my suggestions regarding the Discussion part. The manuscript is now valid for publication. Still, they should consider to provide some English corrections.
